# Consequences of SUR2[A478V] Mutation in Skeletal Muscle of Murine Model of Cantu Syndrome

**DOI:** 10.3390/cells10071791

**Published:** 2021-07-15

**Authors:** Rosa Scala, Fatima Maqoud, Nicola Zizzo, Giuseppe Passantino, Antonietta Mele, Giulia Maria Camerino, Conor McClenaghan, Theresa M. Harter, Colin G. Nichols, Domenico Tricarico

**Affiliations:** 1Section of Pharmacology, Department of Pharmacy-Pharmaceutical Sciences, University of Bari “Aldo Moro”, 70125 Bari, Italy; rosa.scala@uniba.it (R.S.); Fatima.Maqoud@uniba.it (F.M.); antonietta.mele@uniba.it (A.M.); giuliamaria.camerino@uniba.it (G.M.C.); 2Section of Veterinary Pathology and Comparative Oncology, Department of Veterinary Medicine, University of Bari “Aldo Moro”, 70121 Bari, Italy; nicola.zizzo@uniba.it (N.Z.); giuseppe.passantino@uniba.it (G.P.); 3Center for the Investigation of Membrane Excitability Diseases, Department of Cell Biology and Physiology, Washington University School of Medicine, St. Louis, MO 63110-1010, USA; conor.mcclenaghan@wustl.edu (C.M.); harter@wustl.edu (T.M.H.); cnichols@wustl.edu (C.G.N.)

**Keywords:** ATP-sensitive potassium channel, Cantu syndrome, glibenclamide, histopathology, patch-clamp, rare disease, skeletal muscle

## Abstract

(1) Background: Cantu syndrome (CS) arises from gain-of-function (GOF) mutations in the *ABCC9* and *KCNJ8* genes, which encode ATP-sensitive K^+^ (KATP) channel subunits SUR2 and Kir6.1, respectively. Most CS patients have mutations in SUR2, the major component of skeletal muscle KATP, but the consequences of SUR2 GOF in skeletal muscle are unknown. (2) Methods: We performed in vivo and ex vivo characterization of skeletal muscle in heterozygous SUR2[A478V] (SUR2^wt/AV^) and homozygous SUR2[A478V] (SUR2^AV/AV^) CS mice. (3) Results: In SUR2^wt/AV^ and SUR2^AV/AV^ mice, forelimb strength and diaphragm amplitude movement were reduced; muscle echodensity was enhanced. KATP channel currents recorded in Flexor digitorum brevis fibers showed reduced MgATP-sensitivity in SUR2^wt/AV^, dramatically so in SUR2^AV/AV^ mice; IC_50_ for MgATP inhibition of KATP currents were 1.9 ± 0.5 × 10^−5^ M in SUR2^wt/AV^ and 8.6 ± 0.4 × 10^−6^ M in WT mice and was not measurable in SUR2^AV/AV^. A slight rightward shift of sensitivity to inhibition by glibenclamide was detected in SUR2^AV/AV^ mice. Histopathological and qPCR analysis revealed atrophy of soleus and tibialis anterior muscles and up-regulation of atrogin-1 and MuRF1 mRNA in CS mice. (4) Conclusions: SUR2[A478V] “knock-in” mutation in mice impairs KATP channel modulation by MgATP, markedly so in SUR2^AV/AV^, with atrophy and non-inflammatory edema in different skeletal muscle phenotypes.

## 1. Introduction

Cantu syndrome (CS, OMIM#239850) is a rare condition characterized by cardiovascular alterations, skeletal abnormalities and excess hair growth [1], as well as neurological alterations and skeletal muscle dysfunction [2,3]. CS arises from missense mutations in *KCNJ8* or *ABCC9* genes, which encode Kir6.1 and SUR2 subunits, respectively, of ATP-sensitive potassium (KATP) channels. In vitro experiments have revealed that CS mutations are responsible for a gain-of-function (GOF) of the channel, linked to increased open state stability, reduced sensitivity to inhibitory MgATP or increased sensitivity to MgADP [4,5,6,7,8]. Currently, there is no targeted therapy for CS.

Most identified CS mutations are in *ABCC9*; the recently published International Cantu Syndrome Registry reports that 72 of the 74 described patients have confirmed *ABCC9* variants [3]. Several skeletal muscle abnormalities stand out among the large variety of symptoms these patients have reported, including thin or even quite muscular physique during childhood and delays in the development of normal motor skills. Around 70% of these patients have also reported skeletal muscle hypotonia in adulthood, associated with hypermobile joints and exercise intolerance [3]. KATP channels have a key protective role in skeletal muscle, and alteration of their function has previously been related with several pathologic conditions. By opening in response to reduced [ATP]/[ADP] ratio, KATP channels may reduce skeletal muscle excitability and induce fatigue, thereby avoiding exercise-induced muscle damage [9,10]. Consistent with this concept, activation of KATP channels by the agonist pinacidil increases the rate of fatigue in fast- or slow-twitch mice skeletal muscle [11]. Conversely, reducing KATP channels with inhibitory sulfonylureas causes fiber death and atrophy [12,13,14], and down-regulation of WT sarco-KATP channel activity underlies primary and secondary forms of hypokalemic periodic paralysis, transient weakness and hypokalemia [15].

We previously showed that CS mice carrying the GOF Kir6.1 mutation V65M exhibit serious skeletal muscle defects in type I fibers, including atrophy with fibrotic replacement, up-regulation of autophagy genes and reduced muscle strength, in addition to significant cardiovascular pathologies, resembling those seen in human CS [16]. Importantly, while Kir6.1 (and SUR1) contributes to skeletal muscle KATP channels, Kir6.2 pore-forming subunits with SUR2A sulfonylurea receptors are the major components [17]. In the present study, we evaluated the effects induced by a SUR2 CS mutation (A478V) on the skeletal muscle of heterozygous SUR2[A478V] (SUR2^wt/AV^) and homozygous SUR2[A478V] (SUR2^AV/AV^) mice by combined in vivo and ex vivo experiments. Although these mice exhibit less severe vascular defects than Kir6.1[V65M] mice [18,19], the skeletal muscle phenotype was more striking. This is consistent with the more prominent role of SUR2A in generating skeletal muscle channels and suggests that skeletal muscle consequences are likely to be more profound in human CS patients with *ABCC9* versus *KCNJ8* mutations.

## 2. Materials and Methods

### 2.1. Animal Care

CRISPR/Cas9 gene editing was used to introduce single-nucleotide mutations into the endogenous ABCC9 gene locus, resulting in protein substitutions that are analogous to SUR2[A478V] (A476V in mouse sequence) in human CS patients [18]. Heterozygous SUR2[A478V] mice (SUR2^wt/AV^) and homozygous SUR2[A478V] (SUR2^AV/AV^) mice genotype was verified by Sanger sequencing of genomic DNA, as previously described [18]. Since no evidence of gender differences currently exist in CS [3], only male mice were used for our experiments. Pathogen-free mice were imported from Washington University in Saint Louis, under approved protocols at the Stabulario of the Dipartimento di Farmacia-Scienze del Farmaco, University of Bari, Bari, Italy, under the supervision of the veterinary officer in accordance with D.lgs. 26/2014. Mice were maintained 2–4 per cage and were provided with a standard laboratory diet and water ad libitum. Laboratory was kept at 50 ± 5% relative humidity and at a temperature of 22 ± 1 °C, under 12:12 light/dark cycles.

### 2.2. Ethical Statements

All the applied experimental protocols and the animal care were in harmony with the European Directive 2010/63/EU on Animal Protection Used for Scientific Experiments, as well as with the Washington University School of Medicine Institutional Animal Care and use Committee, and were approved by the Italian Ministry of Health and by the Committee of the University of Bari O.P.B.A (Organization for Animal Health) (prot. 8515-X/10, 30 January 2019). The animal sample size (number of mice needed) was calculated as the minimum required to reach statistical significance according to the 3R “Replace, Reduce, Refine” rules. The sample size was calculated using the comparisons between WT mice and CS mice performed by the *t*-test between two independent means (two groups) to evaluate variance between groups. The *p* value was considered statistically significant if lower than 0.05. The calculation was performed according to the approved protocol by our Ethical Committee for animal experiments, O.P.B.A. of the University of Bari. The calculation of the sample was made assuming a delta change of the KATP current amplitude = 300 pA in excised patch experiments performed on FDB fibers of the transgenic mice Kir6.1^wt/VM^ vs. WT mice [16] using G * Power 3.1.9.2. The calculation gave rise to 4 mice x genotype with a power of 0.8.

In particular, in the present work, the primary endpoint used for the calculation of the sample size was the delta changes of the KATP current in excised macropatches. Secondary endpoints were the morphological and functional differences in skeletal muscle.

Assuming the input—tail(s) = two, effect size d = 1.99, α err prob = 0.05, power (1-β err prob) = 0.8, allocation ratio N2/N1 = 1, the mean1 = 1499 pA in FDB for SUR2 homozygous mice, mean2 = 1192 pA in FDB for WT mice in patch-clamp experiments, DS1 = 151, DS2 = 164—the output data were non-centrality parameter δ = 3.079, critical t = 1.8595, df = 8, sample size group 1 = 4, sample size group 2 = 4, actual power = 0.8. We had 4 mice per genotype for a total 8 CS mice and 4 WT mice.

To minimize risk of observer bias and other “experimenter effects”, experiments were performed “blinded” when possible (for instance, homozygous mice are larger and heavier than other genotypes), so that experimenters were unaware of the genotype of the animals [20].

### 2.3. In Vivo Parameters of Muscle Strength

Evaluation of forelimb strength was performed using a grip strength meter (Columbus Instruments, Columbus, OH, USA), according to TREAT–NMD Neuromuscular Network SOPs (SOP ID Number: DMD_M.2.2.001) [21], by an investigator blinded to mouse genotype, as previously described [16]. Five separate measurements were acquired for each animal and were averaged for the calculation of the medium strength.

### 2.4. Ultrasound Evaluations

Blinded ultrasonography experiments were conducted using an ultra-high frequency ultrasound bio-microscopy system Visual Sonic (VEVO 2100, Inc. Toronto, ON, Canada). Each animal was anesthetized via inhalation (induction with 3% isoflurane and 1.5% O_2_ L/min, then constantly maintained via nose cone at 1.5–2% isoflurane and 1.5% O_2_) and placed on a thermostatically controlled table (kept at 37 °C) equipped with 4 copper leads that allowed monitoring of heart and respiratory rate. A rectal probe was used to monitor body temperature. The mice were prepared as previously described [22,23]. We carried out B and M mode diaphragm echodensity measurements using a MS250 probe operating at a frequency of 21 MHz and characterized by lateral and axial resolutions of 165 and 75 mm, respectively. The image acquisition was performed placing the probe along the transverse mid-sternal axis of the mouse. B mode acquisition was used to perform echodensity measurements of the diaphragm and analyzed using ImageJ^®^ software (V1.8.0, Softonic Int., Barcelona, Spain) by creating a grey scale analysis histogram on entire outlined diaphragm sections and on hind limb sections of a constant dimensions of 3653.25 ± 9.75 pixels and of 7392.12 ± 18.45 pixels, respectively. Echodensity differences were expressed as the percentage change of the mean echodensity of all pixels included in the selected area. M mode acquisition was used to measure the amplitude of the diaphragm during normal breathing cycles on the left side of the diaphragm to minimize variability in the measurements. The amplitude of the diaphragm movement during each inspiration (positive deflection) was measured as the distance, in millimeters, between the baseline and the peak of the contraction for each mouse; the diaphragm amplitude movement was obtained from the average of three to five measurements [24].

Ultrasound acquisitions of the hind limb were performed to evaluate total hind limb volume (in mm^3^) and percentage of vascularization (PV%). A 3-dimensional (3D) volume scan of the hind limbs was acquired by translating the ultrasound probe parallel to the long axis of the hind limb. Multiple 2D images were acquired at regular intervals in Power Doppler mode. At the end of the procedure, 3D images were reconstructed from previously collected multiple 2D frames and were visualized with VisualSonics 3D software (VEVO 3100, Toronto, ON, Canada) [25].

### 2.5. Animal Sacrifice and Tissue Collection

Animals were sacrificed by cervical dislocation under ZOLETIL 50/50 (40 mg/kg i.p.) profound anesthesia [26]. Tissues and organs were isolated while keeping the animal under profound anesthesia, using sterilized equipment under a sterile cell culture hood to avoid contamination. Blot-dried organs were weighed; weights were normalized to the tibia length.

### 2.6. Drugs and Solutions

Muscle and organ biopsies were carried out in Ringer’s solution: 145 mM NaCl, 5 mM KCl, 1 mM MgCl_2_, 0.5 mM CaCl_2_, 5 mM glucose and 10 mM 3-(*N*-morpholino) propanesulfonate (MOPS) sodium salt (pH = 7.2 with MOPS acid). For patch-clamp experiments, pipette solution contained 150 mM KCl, 2 mM CaCl_2_ and 1 mM MOPS (pH 7.2); the bath solution contained 150 mM KCl, 5 mM EGTA and 10 mM MOPS (pH 7.2). Stock solution of glibenclamide (Glib) at a concentration of 118.6 mM was prepared by dissolving the drug in dimethyl sulfoxide (DMSO); diluted solutions were prepared using the cell medium or the bath solution, according to the use. DMSO was found not to affect the channel currents or fiber viability at the maximal concentration tested [27].

### 2.7. Patch-Clamp Experiments

Patch-clamp experiments on Flexor digitorum brevis (FDB) fibers were performed in inside-out configuration [28]. Fibers were isolated by enzymatic digestion of the muscle with ~0.5 mg/mL collagenase (C9697, Sigma-Aldrich S.R.L, Milan, Italy). Currents were recorded soon after isolation during voltage steps from a holding potential of 0 mV to −60 mV (Vm) at a 1 kHz sampling rate (filter 0.2 kHz) at room temperature (20–22 °C), using an Axopatch-1D amplifier equipped with a CV-4 head-stage. Current amplitude was measured using Clampfit 10.0 (Molecular Devices, LCC, San Jose, CA, USA). Patch pipettes for excised macropatches were pulled from borosilicate glass capillaries (Glass type 8250, King Precision Glass, Inc. 177 S. Indian Hill Blvd. Claremont, CA, USA) and fire-polished to a final resistance of 1.1 ± 0.1 MΩ. Macropatches containing significant voltage-dependent K^+^ channels or other Kir channels or showing marked loss of channel currents during the time of observation were excluded from the analysis. No correction for liquid junction potential was made, estimated to be <1.9 mV under our experimental conditions. At least 3 FDB muscles per genotypes were used for patch-clamp experiments from different mice.

### 2.8. Histopathological Analysis

An autopsy with macroscopic observation of all organs was performed among the sacrificed subjects according to approved protocols [29]. After animal sacrifice, tissue samples were incorporated, frozen in liquid nitrogen at −80 °C, sectioned using a cryostat for ATPase and succinodehydrogenase (SDH) staining. Other samples were fixed with 10% animal buffered formalin for a minimum period of 72 h. Tissues were embedded in paraffin; sections were cut to 5 μm and stained with standard techniques with hematoxylin and eosin (H.E.), Mallory’s trichrome stain and Schiff’s periodic acid (PAS). Digital images were taken from the cross section at 10–100× magnification to assess muscle fiber morphology and to determine fiber cross-sectional area (CSA) measurements. Images from 20 random fields were acquired for the ten stained sections of each sample using a Leica DMLS D 4000 microscope equipped with an Elements-BR-Nikon NIS camera and image analyzer. Section analysis and CSA evaluation of the fibers were performed by QWin software (3.1.0, Leica, Wetzlar, Germany) on 53 cells per muscle section. Examination of cell morphology and intracellular structures was conducted, and the severity of lesions observed was assessed [30] by an animal pathologist blinded to the mouse genotype. At least 2 SOL, 2 TA, 2 GA and 2 EDL muscles for histopathology analysis were used from 2 different mice per genotype.

### 2.9. Isolation of Total RNA, Reverse Transcription and PCR

Total RNA was isolated and purified from entire Flexor digitorum brevis (FDB), soleus (SOL), gastrocnemius (GA) and tibialis anterior (TA) muscles with Trizol reagent (Invitrogen Life Technologies, Waltham, MA, USA), and was quantified using a spectrophotometer (ND-1000 Nano-Drop, Thermo Fisher Scientific Waltham, MA, USA) For evaluating KATP channel composition, PCR amplification was performed using PCR Master Mix (Promega Italia S.r.l. Soc. Unip., Milan, Italy). PCR cycles consisted of denaturation at 95 °C for 1 min, annealing segment at 58 °C for 1 min, and extension at 72 °C for 1 min, repeated for 30 cycles. Primer sequences were previously reported [16]. Amplified PCR products were finally separated on 1% agarose gel. For investigating muscle damage, qPCR analysis was performed as previously described [12,23]. Probes (Life Technologies, Monza, Italy) were ordered with the following assay IDs: *FBXO32* Mm00499523_m1, *TRIM63* Mm01185221_m1; *Actb* Mm00607939_s1. Data collection and analysis were performed according to the MIQE [31]. At least 3 sample muscles from 3 different mice per genotype were used.

### 2.10. Fiber Survival Evaluation

Survival analysis for FDB fibers was performed by an investigator blinded to mouse genotype and treatment. Before the analysis, isolated fibers were seeded and equilibrated in DMEM supplemented with 10% fetal bovine serum, 1% L-glutamine and 1% penicillin-streptomycin at 37 °C for at least 30 min [13]. Fibers were monitored for changes in morphological parameters such as length and diameter within 24 h after excision, using a Nikon TMS Inverted Microscope 4× magnification. Changes of ≥40% in morphological parameters defined dead fibers, and the appearance of multiple sarcolemma blebs was considered a prognostic sign of cellular death. At least 3 FDB muscles per genotype from different mice were used for cell count.

### 2.11. Data Analysis and Statistics

Data were collected and analyzed using Excel software (Microsoft Office 2010, Innovation Campus, Milan, Italy), Clampfit 10.5 (Molecular Devices) and SigmaPlot 10.0 (Systat Software, San Jose, CA, USA). Results are presented as mean ± SEM unless otherwise indicated. The number of replicates relative to each experimental dataset is reported in the figure description. One-way analysis of variance (ANOVA) followed by multiple comparisons tests were applied (*p* < 0.05 unless otherwise indicated). Student *t* test was applied for comparison of significance between means. Data were considered significantly different for *p* < 0.05 unless otherwise specified.

For patch-clamp experiments, the percentage of KATP current inhibition induced by glibenclamide (Glib) and MgATP was calculated with the following equation: −(I_CTRL_ − I_drug_)/(I_CTRL_ − I_leak_) × 100, where I leak was the current recorded after the application of 5 × 10^−3^ M MgATP on excised macropatches from skeletal muscle fibers.

## 3. Results

### 3.1. SUR2 CS Mice Are Heavier Than WT but Have Weaker Muscles

We first assessed the macroscopic properties of skeletal muscle in terms of strength and integrity. Ambulatory behavior and gross musculoskeletal phenotype of CS mice were not obviously different from wild type (WT), although 33-week-old SUR2^wt/AV^ and SUR2^AV/AV^ mice were ~10% heavier than age-matched WT mice (SUR2^wt/AV^ mice = 35.17 ± 2.21 g (student *t* test *p* < 0.05); SUR2^AV/AV^ mice = 33.5 ± 2.81 g (student *t* test *p* > 0.05, n.s.); WT mice = 30.33 ± 1.25 g (*n*. mice per genotype = 4), while forelimb strength evaluation showed that both SUR2^wt/AV^ and SUR2^AV/AV^ mice were significantly (~10%) weaker than WT (Figure 1A).

### 3.2. Skeletal Muscles of SUR2 Mutated Mice Show Morphological and Functional Abnormalities

A 3D ultrasound evaluation of hind limb was performed to assess the morphological differences in skeletal muscle structure. There was a progressive increase in hind limb echodensity above that in WT, from ~20% in SUR2^wt/AV^ mice to ~40% in SUR2^AV/AV^ mice (Figure 1B), indicating progressively lower muscle mass and fibrotic tissue/fat deposition with increasing gene dose. The increased echodensity in SUR2^wt/AV^ was paralleled by increased hind limb volume (Figure 1D), without change in vascularization (Figure 1C). In contrast, in SUR2^AV/AV^ mice, hind limb volume was not markedly different from WT (Figure 1B), but there was a striking (~60%) increase in vascularization (Figure 1C). Ultrasound evaluation of the diaphragm revealed significant ~35% reduction in the amplitude of diaphragm movement in both SUR2^wt/AV^ and SUR2^AV/AV^ mice, associated with ~20% and ~40% enhancement of echodensity, respectively (Figure 1E,F), again indicative of progressively worse fat/fibrotic deposition.

Following sacrifice, individual muscles were isolated, blotted and weighed. Increased muscle weights were observed in the heart, pectoral muscle, tibialis anterior (TA) and flexor digitorum brevis (FDB) from SUR2^wt/AV^ and more so from SUR2^AV/AV^ mice. In contrast, weight was reduced in soleus (SOL) and (non-significantly) in extensor digitorum longus (EDL) muscles from these mice (Appendix A).

### 3.3. KATP Channels in SUR2 CS Muscle Fibers Show Reduced Sensitivity to MgATP

Expression of KATP channel subunits was confirmed in fast-twitch FDB fibers by PCR analysis of *KCNJ8*, *KCNJ11*, *ABCC8* and *ABBC9* genes (encoding Kir6.1, Kir6.2, SUR1 and SUR2, respectively), from FDB (*n*. muscles/mice = 2/2) whole muscle samples. The presence of specific bands for *KCNJ11* and *ABCC9* (Figure 2A) was consistent with the prevailing evidence that skeletal muscle KATP channels are composed predominantly of Kir6.2/SUR2 subunits [17]. KATP channel activity was recorded in excised macropatches using the patch-clamp technique in acutely dissociated fast-twitch FDB fibers from WT, SUR2^wt/AV^ and SUR2^AV/AV^ mice. In 0 ATP bath solution, KATP current amplitude in FDB fibers from CS mice was not significantly different from WT (Figure 2B,C). We next assessed the nucleotide regulation of native KATP channels in FDB skeletal muscle fibers. There was significantly reduced sensitivity to inhibition by MgATP in SUR2^wt/AV^ fibers and more marked reduction in SUR2^AV/AV^ FDB fibers (Figure 3A–C, Table 1), consistent with the loss of MgATP sensitivity expected as the mechanism for increased activity of this mutation, based on recombinant channel activity recorded in cell lines [6].

In addition, we tested sensitivity to the KATP channel blocker glibenclamide (Glib). Glib sensitivity was slightly decreased in channels from SUR2^wt/AV^ FDB fibers and more significantly so in SUR2^AV/AV^ (Figure 3B–D, Table 1), again reflecting the minimal effect of the mutation on glibenclamide sensitivity in recombinant channels and in other tissues [6,18].

### 3.4. Marked Atrophy and Abnormal Morphology of SOL and TA Muscles

Histochemical evaluations were performed on slow-twitch soleus (SOL) (*n*. muscles ≥ 2 from ≥2 mice per genotype), fast-twitch gastrocnemius (GA) (*n*. muscles ≥ 2 from ≥2 mice per genotype), tibialis anterior (TA) (*n*. muscles ≥ 2 from ≥2 mice per genotype) and extensor digitorum longus (EDL) (*n*. muscles ≥ 2 from ≥2 mice per genotype) muscles. Hematoxylin–eosin (H.E.) stains indicated non-inflammatory edema in the endomysia of GA and EDL muscles of SUR2^AV/AV^ and SUR2^wt/AV^ mice. The cross-sectional area (CSA) of EDL (Figure 4A–C) and GA muscles (data not shown) did not differ significantly between genotypes as evaluated by one-way ANOVA (*p* > 0.05). In contrast, CSA was progressively reduced, by ~5% and ~20% in SOL muscle from SUR2^wt/AV^ and SUR2^AV/AV^ mice, respectively (Figure 5A–D), and this was associated with relatively small diameter myofibers and the presence of inflammatory cells (*n*. inflammatory cells per sections = 15 ± 3, 20×) in the muscle of SUR2^AV/AV^ mice, as well as some degenerative-necrotic areas (Figure 5D). SOL muscle section of WT mice showed *n*. 9 ± 3 inflammatory cells per sections, 20×.

TA muscle showed even more severe atrophy, with reduced CSA of ~30% and ~40% in TA of SUR2^wt/AV^ and SUR2^AV/AV^ mice, respectively (Figure 6A–C, Figure 7). In the muscle TA of SUR2^AV/AV^ mice, a heterogeneous population of myofibers with reduced area was evident, with angular atrophic fibers alongside larger, rounded and slightly hypertrophic fibers (Figure 6D,E), as well as interstices with edema and inflammatory cells (Figure 6E) (*n*. inflammatory cells per sections = 18 ± 4, 20×). TA muscle section of WT mice showed *n*. 7 ± 4 inflammatory cells per sections, 20×.

Mallory trichrome and P.A.S. analysis with blue and magenta red color were negative in all sampled muscles, including TA and SOL, suggesting the absence of connective tissues and glycogen accumulation, respectively, in the CS muscles. Additionally, the ATP-ASE reaction to pH 4.3 and coloring of the myofibrils positive to the succinodehydrogenase (SDH) was negative in the muscles analyzed, suggesting the lack of involvement of type I fibers (*n*. muscles ≥ 2 from ≥2 mice per genotype).

In support to histological analysis, qPCR showed the up-regulation of *FBXO32* and *TRIM63* (encoding atrogin-1 and MuRF1, respectively) in SOL and TA muscles from SUR2^wt/AV^ and in SUR2^AV/AV^ mice with respect to the WT (Figure 8). Furthermore, consistent with non-apoptotic death, no expression of Casp3 was detected in either WT, SUR2^wt/AV^ or SUR2^AV/AV^ muscles (data not shown).

### 3.5. Ex Vivo Survival of Skeletal Muscle Fibers from SUR2 Mutated Mice Is Reduced

Finally, we assessed the ex vivo survival of isolated FDB fibers in the three genotypes. There was no marked difference in FDB survival rate within 3 h from the excision, but there was markedly higher death of SUR2^AV/AV^ FDB fibers within 24 h of excision, with a viability reduction of ~30% with respect to the other two phenotypes. Consistent with a pre-existing fiber weakness, inhibition of KATP by application of Glib did not strongly affect the survival rate of WT and SUR2^wt/AV^ fibers in short term incubation, instead slightly worsening fiber survival in all the genotypes after 24 h of incubation (Figure 9).

## 4. Discussion

In the present work we investigated the effects of the patient-specific Cantu syndrome (CS) *ABCC9* mutation SUR2[A478V] in different skeletal muscle types from SUR2^wt/AV^ and SUR2^AV/AV^ CS mice, in which mutation was introduced to the equivalent locus in the mouse genome [18]. Although CS mice were heavier than age-matched WT mice, we found markedly impaired strength and muscle function in heterozygous, and more so in homozygous, CS mice, similar to what is reported in human CS patients [3]. CS mice generate significantly lower forelimb forces than WT, similar to the reduced force previously observed in Kir6.1[V65M] mice [16,32], which carry a Kir6.1 CS mutation.

### 4.1. Skeletal Muscle Weakness in CS

A variety of muscles, including SOL, TA, FDB and diaphragm, with different physiological roles, were similarly affected by the mutation, all showing atrophy and low fiber survival ex vivo that contribute to the characteristic muscle phenotypes in vivo. Large inter-myofiber areas appear to be formed by edema of not-inflammatory or lymphatic origin, being negative to Mallory trichrome or markers of fibrosis and showing the presence of inflammatory infiltrate. These histological changes, together with reduced muscle cross-sectional area (CSA) and weight in SOL, and of CSA in TA, help to explain muscle weakness yet enhanced hind-limb volume and mean echodensity evaluated by 3D ultrasonography. Reduction of CSA and muscle weights in SOL but only of CSA in TA could be explained by the deposition of interstitial material that masks the expected reduction of muscle weight. Interstitial fat deposition cannot be excluded as a possible causative factor affecting muscle function and morphology since the use of ethanol solvent in our histopathological analysis excludes identification of adipose cells, which could potentially infiltrate the muscle. We also observed a large reduction of the amplitude of diaphragm movement of SUR2^wt/AV^ and SUR2^AV/AV^ mice, consistent with exercise intolerance and difficulty breathing that has been reported in CS patients [3]; in our mice this was also associated with enhanced diaphragm echodensity. GA and EDL muscles were much less affected, with no change in CSA or muscle weight, despite the presence of some edema of non-inflammatory origin. This suggests that not all muscles are equally affected, and potentially that the specific muscle function plays a role in determining the CS muscular phenotype.

Atrophy with reduction of CSA was observed in SOL and TA muscles, and reduced ex vivo survival was observed in hetero and homozygous SUR2 FDB fibers. Interestingly, this was not reversed or prevented by KATP channel inhibition by glibenclamide treatment in vitro, suggesting a pre-existing muscle deficit.

### 4.2. The Cellular and Pharmacological Consequences in SUR2^wt/AV^ and SUR2^AV/AV^ Mice Muscles

Excised patch-clamp experiments revealed no significant increase of KATP current density in FDB fibers of SUR2^wt/AV^ and SUR2^AV/AV^ CS mice but did indicate a significant reduction of the sensitivity to inhibition by MgATP in the heterozygous and even more so in the homozygous mice, which was at least as marked as previously seen in cardiac myocytes [18], together with a minimal shift in glibenclamide sensitivity. Previously, we showed only a small shift of the MgATP sensitivity in FDB and SOL patches from Kir6.1^wt/VM^ fibers, with a reduced response to Glib in FDB patches and loss of response in SOL patches expressing the Kir6.1 subunit [16]. These findings are consistent with the prominence of Kir6.2 and SUR2A (but not Kir6.1) in skeletal muscle KATP channels function, a point we return to below.

### 4.3. Mechanistic Basis of CS Skeletal Muscle Pathology

KATP channels are abundant in cardiac, skeletal and vascular smooth muscle [33]. Skeletal muscle KATP channels are expected to open as [ATP]/[ADP] falls during repetitive contractions, leading to hyperpolarization and action potential failure, which may limit contractile force. All four KATP channel subunits are expressed in skeletal muscle, but the major subunits are understood to be Kir6.2 (*KCNJ11*) and SUR2A (*ABCC9*) [17]. Kir6.2 knockout mice reveal rapidly fatiguing, weak muscles [34], indicating that the failure of KATP to limit excitability results in fatigue and ultimately necrosis [35] as a consequence of [Ca]-overload and energy depletion [36].

The underlying KATP channel genes are present as two distinct pairs (*ABCC8/KCNJ11* and *ABCC9/KCNJ8*), on human chromosomes 11p and 12p, respectively. They are generally expressed as these canonical pairs in specific tissues, and distinct human syndromes are associated with mutations in each pair of genes; gain- (GOF) or loss-of-function (LOF) mutations in either SUR1 or Kir6.2 cause congenital or neonatal diabetes, respectively [37]. Conversely, GOF mutations in either Kir6.1 or SUR2 cause Cantu syndrome (CS), with quite different organ involvement [2,3,18]. The expression of non-canonical KATP channels formed by Kir6.2 with SUR2A in skeletal and cardiac muscle would suggest the potential for overlap of neonatal diabetes and Cantu syndrome pathologies in these tissues. However, there is essentially no cardiac and skeletal muscle phenotype in NDM patients [38], despite reported altered muscle function [39], or in mice expressing Kir6.2 GOF specifically in the heart [40]. There is a striking cardiac phenotype in CS (massive enlargement and hypercontractility) [18], but this is a secondary consequence of neuro-humoral feedback in response to the reduced systemic vascular resistance that results from overactive Kir6.1/SUR2B channels in blood vessels [19].

SUR2A is the main KATP channel regulatory subunit in all skeletal types, and marked changes in KATP currents are present in the SUR2[A478V] CS skeletal myocytes, potentially underlying the pathologies observed. SUR2A and Kir6.2 subunits are prominently expressed in TA and FDB WT rat muscles, generating a high KATP channel current density in these muscles [17], thereby potentially increasing their susceptibility to the metabolic uncoupling associated with GOF SUR2 CS mutations. SOL muscle is an oxidative muscle that is more susceptible to metabolic stress than glycolytic muscles such as GA and EDL, which are expected to face metabolic stress more efficiently.

However, contribution of non-skeletal myocyte mechanisms to skeletal muscle pathology is also possible. In this context, significant vascularization is observed in the CS muscles, especially in homozygous SUR2^AV/AV^ mice, and the potential involvement of vascular or other smooth muscle derangements in the skeletal muscle phenotypes should be considered. Previously, even though Kir6.1 is only a minor component of skeletal muscle KATP, we showed that muscles from Kir6.1^wt/VM^ CS mice also tended to be heavier than WT, with significant increase of echodensity of hind limb muscles but also associated with a tendency toward increased hind limb volume and mild extra-vascularization. Additionally, histological analysis revealed marked atrophy of type I fibers with extended necrotic area and replacement of the myofibers, suggesting damage to skeletal muscle integrity, along with fibrous tissue deposition and infiltration of inflammatory cells, explaining increased echodensity, decreased muscle strength and pseudo-hypertrophy [16]. As discussed above, and consistent with only a minor role for Kir6.1 in skeletal muscle channels, the skeletal muscle KATP GOF, as revealed by shifts in ATP sensitivity, were much less than in the present study, even though the muscle pathology was similar.

This is reminiscent of parallel findings in the heart in which cardiac myocyte KATP GOF is present in SUR2^AV/AV^ but not Kir6.1^wt/VM^, yet pathologic enlargement and hypercontractility is greater in the latter [18]. It is explained by up-regulation of the renin–angiotensin signaling pathway as a major mediator of feedback signaling from lowered resistance in the systemic vasculature to drive cardiac enlargement and hypercontractility [41]. Thus, although overactivity of skeletal myocyte KATP channels might contribute to atrophy, ACE/Ang II/ATR1 signaling is also a known mediator of atrophy and sarcopenia in skeletal muscle [42] and a potential major driver of the muscle pathology. Similarly, fiber damage may result in fluid leakage into the extracellular space, but the nature of the observed fluid, combined with the demonstration that lymphatic contractility can be markedly reduced by KATP GOF in lymphatic smooth muscle [43], further suggests a non-skeletal myocyte, i.e., lymphatic smooth muscle [43], contributor to the edema that characterizes the CS muscles.

This work aimed to investigate the KATP channel functionality in skeletal muscle of CS mice so that the low number of mice limited the investigation of secondary endpoints. Future investigations having as a primary endpoint the evaluation of the molecular pathways associated with this dysfunction will help to clarify this point.

## 5. Conclusions

The introduction of CS mutations on SUR2^wt/AV^ and SUR2^AV/AV^ mice, which strongly reduces the sensitivity of skeletal muscle fibers to ATP inhibition, has marked effects on the integrity and the functionality of skeletal muscles, leading to muscle weakness and atrophy, paralleling findings in human CS patients. This and other SUR2 mutations do not markedly affect the channel sensitivity to glibenclamide. Thus, as has been shown for cardiovascular pathologies [19,41], glibenclamide treatment may help to reverse or avoid these consequences.

## Figures and Tables

**Figure 1 cells-10-01791-f001:**
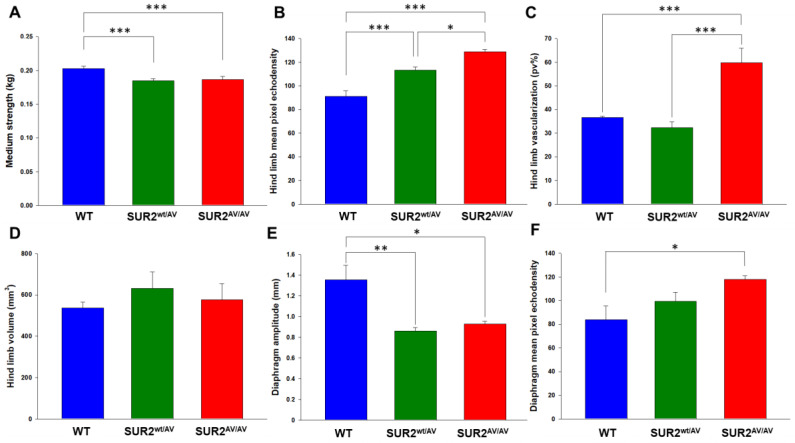
In vivo evaluation of skeletal muscle morphology and functionality (*n*. animal used for these evaluations: 4 animals per genotype). (**A**) Medium forelimb strength was 0.203 ± 0.003 kg in WT mice (*n*. measurements = 16), 0.185 ± 0.003 in SUR2^wt/AV^ kg mice (*n*. measurements = 21) and 0.191 ± 0.005 kg in SUR2^AV/AV^ mice (*n*. measurements = 20), respectively. In ultrasound evaluation, (**B**) the mean pixel echodensity for the hind limb was 91.2 ± 4.7 in WT mice, 113.4 ± 2.4 in SUR2^wt/AV^ mice and 128.7 ± 2 in SUR2^AV/AV^ mice. (**C**) Percentage of hind limb vascularization was 36.6 ± 0.6% in WT mice, 32.3 ± 2.5% in SUR2^wt/AV^ mice and 59.8 ± 6.2% in SUR2^AV/AV^ mice. (**D**) Hind limb volume was 536.7 ± 29.2 mm^3^ in WT mice, 630.8 ± 79.9 mm^3^ in SUR2^wt/AV^ mice and 575.8 ± 77.8 mm^3^ in SUR2^AV/AV^ mice. (**E**) Diaphragm amplitude was 1.35 ± 0.14 mm in WT mice, 0.86 ± 0.04 mm in SUR2^wt/AV^ mice and 0.93 ± 0.03 mm in SUR2^AV/AV^ mice. (**F**) The mean pixel echodensity for the diaphragm was 84 ± 11.4 in WT mice, 99.3 ± 7.5 in SUR2^wt/AV^ mice and 118 ± 2.8 in SUR2^AV/AV^ mice. Data are presented as mean ± SEM. Data significantly different as evaluated with one-way ANOVA and Bonferroni test (*** *p* < 0.001, ** *p* < 0.005, * *p* < 0.05).

**Figure 2 cells-10-01791-f002:**
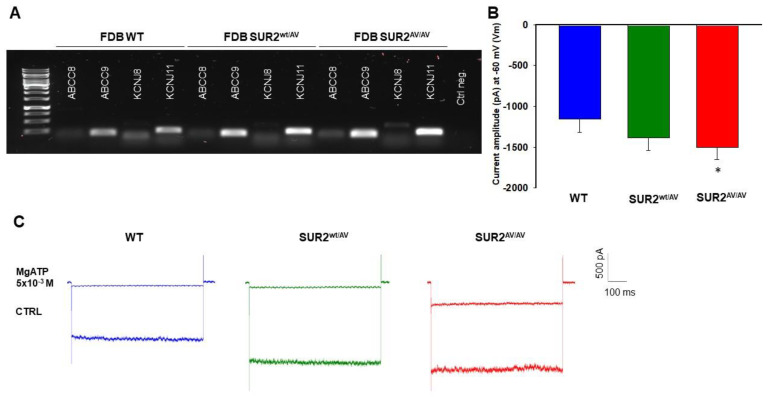
Channel subunits expression and current amplitude distribution in FDB fibers at physiological membrane voltage of −60 mV (Vm). (**A**) Sample gel of PCR analysis in an FDB muscle: high levels of expression of *ABCC9* and *KCNJ11* genes, encoding respectively for SUR2 and Kir6.2, were found in all the phenotypes. (**B**) In inside-out patch-clamp experiments on FDB isolated fibers, the calculated mean was −1152.8 ± 164 pA for WT, −1381.2 ± 155.5 pA for SUR2^wt/AV^ mice and −1499.2 ± 151.7 pA for SUR2^AV/AV^ mice (*n*. patches = 20, 38, 45 from 3 muscles/3 mice per genotype, respectively); * data significantly different as evaluated with one-way ANOVA and Bonferroni test for *p* < 0.05. (**C**) Sample traces of recorded KATP currents, illustrating a tendency toward increased current amplitude in FDB fibers from SUR2^wt/AV^ and SUR2^AV/AV^ mice with respect to WT but more marked reduction of sensitivity to 5 × 10^−3^ M MgATP.

**Figure 3 cells-10-01791-f003:**
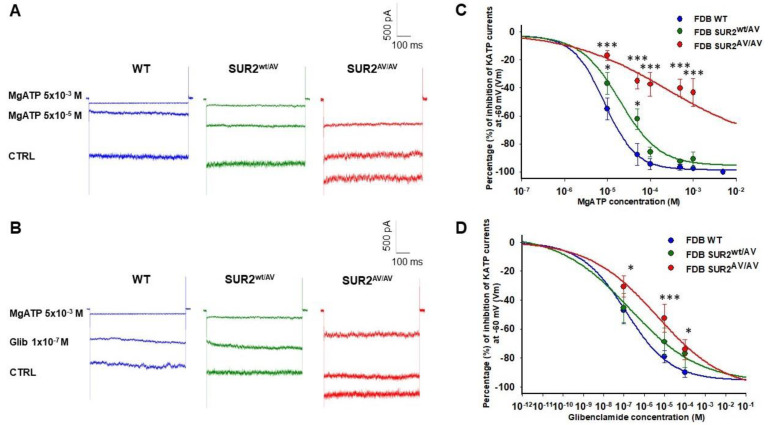
MgATP- and glibenclamide-sensitivity of KATP currents in FDB fibers. (**A**) Sample traces of currents in zero (CTRL), 5 × 10^−5^ M and 5 × 10^−3^ M MgATP in FDB isolated fibers from WT, SUR2^wt/AV^ and SUR2^AV/AV^ mice. (**B**) Sample traces of currents in zero (CTRL), 10^−7^ M glibenclamide (Glib) and 5 × 10^−3^ M MgATP in FDB fibers from WT, SUR2^wt/AV^ and SUR2^AV/AV^ mice (*n*. patches = 30, 48, 55 from 3 muscles/3 mice per genotype, respectively). (**C**,**D**) Sensitivity to inhibition by MgATP (**C**) or Glib (**D**) in FDB patches from WT, SUR2^wt/AV^ and SUR2^AV/AV^ mice. Data were fitted using the Hill equation (see Table 1). Each experimental point represents the mean ± SEM of at least three patches. Data significantly different as evaluated with one-way ANOVA and Bonferroni test (*** *p* < 0.001, * *p* < 0.05).

**Figure 4 cells-10-01791-f004:**
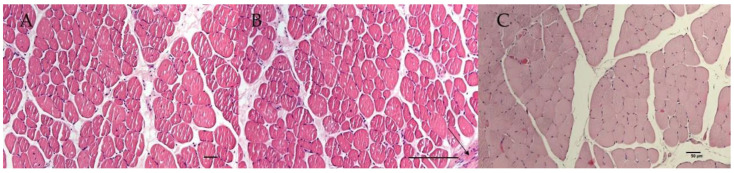
Sample representative histological analysis with hematoxylin–eosin (H.E.) reaction on extensor digitorum longus (EDL) muscles of (**A**) WT, (**B**) SUR2^AV/AV^ and SUR2^wt/AV^ mice. (**B**) EDL muscle of homozygous CS mice showed homogeneous myofibers population and normal nuclei, except for pink colored materials in the endomysia and perimysium, suggestive of edema of non-inflammatory origin (arrow) and no atrophy (H.E. 10×, Bar 50 μm). Heterozygous (**C**) and WT (**A**) EDL muscles sections were normal.

**Figure 5 cells-10-01791-f005:**
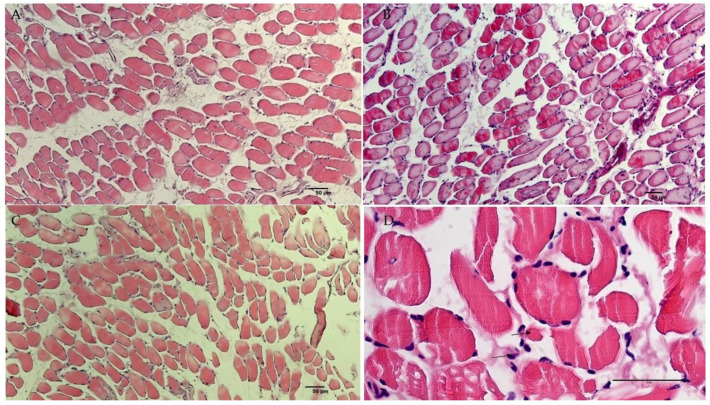
Sample representative histological analysis with hematoxylin–eosin (H.E.) reaction on soleus (SOL) muscles of (**A**) WT, (**B**,**D**) SUR2^AV/AV^ and (**C**) SUR2^wt/AV^ mice. (**B**) SOL muscles from SUR2^AV/AV^ mice showed marked atrophy (H.E. 10×, Bar 50 μm) vs. (**A**) WT and (**C**) SUR2^wt/AV^ EDL muscles. (**D**) SOL muscle from SUR2^AV/AV^ mice showed myofibers with small diameter (arrow 1) and inflammatory cells (arrow 2) and rare area of degeneration and necrosis with some atrophy (H.E. 20×, Bar 50 μm).

**Figure 6 cells-10-01791-f006:**
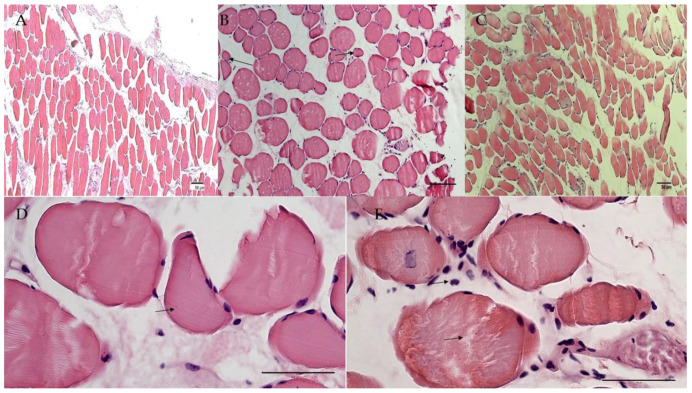
Sample representative histological analysis with hematoxylin–eosin (H.E.) reaction on tibialis anterioris (TA) muscles of (**A**) WT, (**B**,**D**,**E**) SUR2^AV/AV^ and (**C**) SUR2^wt/AV^ mice. (**B**) TA muscle from SUR2^AV/AV^ mice showed heterogenous population of myofibers with reduced area (arrows) (10×, Bar 50 μm), (**D**) no longer polygonal in shape but elongated, rounded and angular, with hyper-eosinophil sarcoplasm (arrow) (H.E. 40×, Bar 50 μm). (**E**) Hyperchromatic nuclei can be grouped together, and present fragmented myo-fibers separated by an accumulation of pale pink interstitial material of inflammatory origin in TA muscle from SUR2^AV/AV^ mice (H.E. 40×, Bar 50 μm).

**Figure 7 cells-10-01791-f007:**
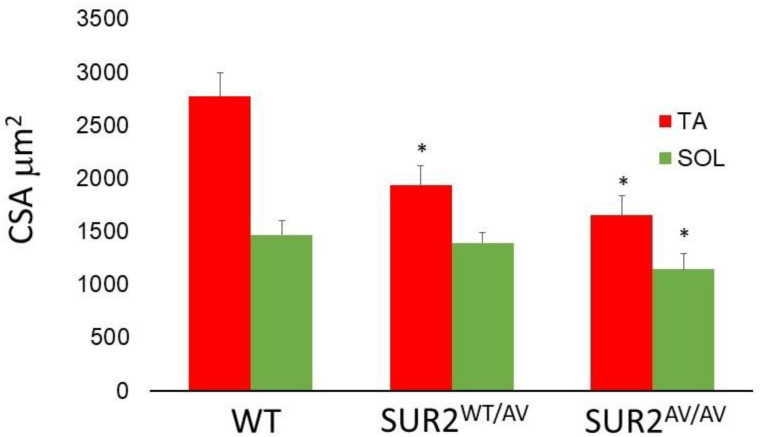
Cross sectional area of tibialis anterior (TA) and soleus (SOL) muscles of CS mice. The CSA of TA muscle was significantly reduced in CS mice, and it was 1937.5 ± 121 μm^2^ in heterozygous (F = 2.982, * *p* < 0.05) (*n* muscle/mice = 3/3) and 1647.8 ± 131 μm^2^ in homozygous CS mice (F = 3.321, * *p* < 0.05) (*n* muscle/mice = 3/3) vs. 2769.6 ± 221 μm^2^ in WT mice (*n* muscle/mice = 3/3). The CSA of SOL muscle was significantly reduced in homozygous mice, and it was 1142.7 ± 99 μm^2^ (F = 1.9821, * *p* < 0.05) (*N* mice = 2) vs. 1465 ± 91 μm^2^ in WT (*n* muscle/mice = 3/3). CSA was calculated on 53 fibers per mouse.

**Figure 8 cells-10-01791-f008:**
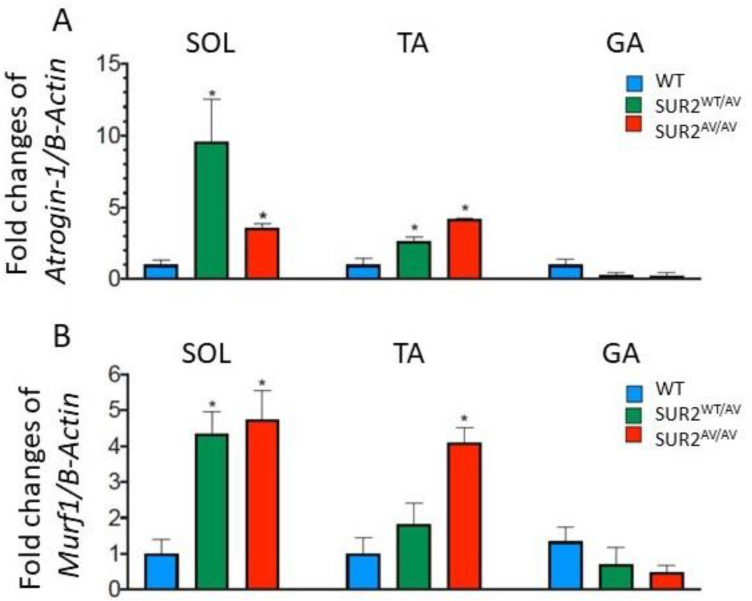
Atrogin-1 and MuRF1 gene expression in soleus (SOL), tibialis anterior (TA) and gastrocnemius (GA) of WT, SUR2^wt/AV^ and SUR2^AV/AV^ mice. The figure shows quantification of transcriptional level with qPCR for (**A**) atrogin-1 and (**B**) MuRF1 normalized by β-actin gene. In each graph, bars represent mean ± SEM for 3 muscles per each genotype. * Data significantly different with respect to the WT (*p* < 0.05).

**Figure 9 cells-10-01791-f009:**
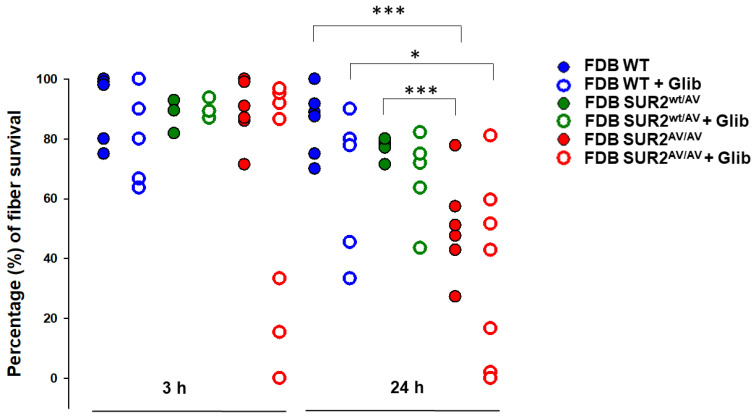
Evaluation of ex vivo survival of FDB isolated fibers from WT, SUR2^wt/AV^ and SUR2^AV/AV^ mice. Fiber survival was reduced at 3 h in SUR2^AV/AV^ fibers and progressively more so in SUR2^wt/AV^ and SUR2^AV/AV^ at 24 h. Short-time incubation with Glib did not significantly affect fiber viability, whereas 24 h of fibers treatment with Glib further reduced fiber viability with respect to untreated fibers (*n*. of fibers = 100–150 for each experimental condition; fibers were collected from ≥ 3 FDB from ≥3 mice per genotype). Values of percentage of fiber survival are presented as individual data points for each repetition. Data significantly different as evaluated with one-way ANOVA and Bonferroni test (*** *p* < 0.001, * *p* < 0.05).

**Table 1 cells-10-01791-t001:** Fitting parameters measured the concentration–response relationships of KATP currents amplitude vs. MgATP or glibenclamide concentrations in FDB isolated fibers from WT, SUR2^wt/AV^ and SUR2^AV/AV^ mice. Values are expressed as the mean ± SEM of at least three replicates from 3 muscles/3 mice per genotype, as evaluated by using SigmaPlot 10. n.a. indicates experimental conditions for which fitting parameters were not obtained.

		MgATP			Glibenclamide	
Mice	Emax (%)	IC_50_ (M)	Hill Slope	Emax (%)	IC_50_ (M)	Hill Slope
WT	−98.7 ± 0.7	8.6 ± 0.4 × 10^−6^	1.2 ± 0.1	−95.6 ± 1.9	1.2 ± 0.4 × 10^−7^ M	0.4 ± 0.05
SUR2^WT/AV^	−95.6 ± 6	1.9 ± 0.5 × 10^−5^	0.9 ± 0.2	−97 ± 7.5	1.9 ± 2.2 × 10^−7^ M	0.25 ± 0.08
SUR2^AV/AV^	n.a	n.a.	n.a.	−102.5 ± 9	4.2 ± 3.7 × 10^−6^ M	0.26 ± 0.07

## Data Availability

The raw data supporting the conclusions of this article will be made available by the authors, without undue reservation, to any qualified researcher.

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
