# Peer review of "Consequences of SUR2[A478V] Mutation in Skeletal Muscle of Murine Model of Cantu Syndrome"

_cells, 2021, doi:10.3390/cells10071791_

Round 1

Reviewer 1 Report

My previous concerns have been adequately addressed.

Author Response

The authors would like to thank the reviewer for his/her positive comments on our manuscript.

Reviewer 2 Report

Scala R and collaborators adopted several suggestions made at the previous review. However, smaple size per analysis is still not clear. I also have some minor comments.

Line 182 – Microfreezer would it be a cryostat?

Line 203 and 226 – I am not sure about this: histochemistry and PCR analysis were performed in two animals only?

“All experiments were performed in duplicates per muscle and in duplicates per genotypes”.

Duplicate would be for the same sample (sorry if I misunderstood, but it is confusing).

Line 22 – “Student t test was applied for comparison of significance between mean for p<0.05.” this phrase needs to be rewritten.

Why Student’s T test were performed to compare muscles weight?

Table 1, figure 4, 5 – how many animals per group? This information should be mentioned in a standardized form in all figures and table.

Figure 8 and for all the PCR analysis: would be more informative to have a graphic or table with the mean of the gene evaluated in addition to the gels.

Author Response

Scala R and collaborators adopted several suggestions made at the previous review. However, the sample size per analysis is still not clear. I also have some minor comments.

Line 182 – Microfreezer would it be a cryostat?

OK

Line 203 and 226 – I am not sure about this: histochemistry and PCR analysis were performed in two animals only?

“All experiments were performed in duplicates per muscle and in duplicates per genotypes”.

The duplicate would be for the same sample (sorry if I misunderstood, but it is confusing).

Author: The reviewer is right the terminology was confusing, the experiments are performed in at least 2 muscles per genotype, sometimes 3 when indicated.

PCR and histological data (cell count in case of FDB) were collected from at least 2 SOL, 2 TA, 2 GA, 2 FDB and 2 EDL muscles for PCR and 2 SOL, 2 TA, 2 GA, and 2 EDL muscles for histopathology, and 3 FDB muscles for cell count for a total of 4 muscles per genotype. In the case of confirmatory analysis, spare muscles were used, indeed we lost some samples for technical problems during PCR (too low mRNA extraction), failure of histopathological protocols, or the low number of enzymatically isolated FDB fibers. 3 FDB muscles per genotype were used for patch-clamp experiments.

Line 22 – “Student t-test was applied for comparison of significance between mean for p<0.05.” this phrase needs to be rewritten.

Why Student’s t-test was performed to compare muscle weight?

Author: The reviewer is right one WAY ANOVA was used instead

Table 1, figure 4, 5 – how many animals per group? This information should be mentioned in a standardized form in all figures and tables.

Author: the number of mice was 4 per genotype

Figure 8 and for all the PCR analysis: would be more informative to have a graphic or table with the mean of the gene evaluated in addition to the gels.

We agree with the reviewer about the need for more quantitative analysis. The sample gel reported in figure 8 is however representative of at least 2 gel performed in two different mice with the same results. We should clarify that the primary goal of our manuscript is the evaluation of KATP channel activity and delta change of KATP currents in excised macro-patch experiments and response to endogenous modulators such as ATPMg to evaluate metabolic decoupling in homo and heterozygous SUR2CS mice, and the response to glibenclamide in vitro. To reduce the number of mice to a minimum according to the 3R requirement, the sample size was therefore calculated on the delta change of patch-clamp currents recorded in WT and CS homo mice, and this calculation results in 4 mice x genotype for a total of 8 CS mice and 4 related WT mice. So we had 8 Soleus, FDB, TA, EDL, and GA muscles x genotype.  The secondary goal was the evaluation of gene expression looking for on/off changes and histopathological changes in these mice. So the sample size was calculated on the primary endpoint as required from our previously approved Ethical protocol. This is why we do not have so many mice for more extensive investigations.

Round 2

Reviewer 2 Report

Two muscle per animals should be considered duplicates and not a new sample.

The low number of mice is a great concern, as authors cannot make any assumptions without a statistical analysis. In this case, I would like to suggest that data from PCR and histological analysis of the two animals per group should be presented.

Authors must address this limitation in the discussion.

Author Response

We thank the reviewer for the helpful comments

The reviewer is right. For the PCR and some histological analysis, we used at least two muscles coming from at least two mice for each genotype. This is now reported in the figure legends and in the text as follows: n. muscles> 2 from >2 mice per genotype.

We specified in the text that in the present work, the primary endpoint used for the calculation of the sample size is the delta changes of the KATP current in excised macro patches. Secondary endpoints are the morphological and functional differences in skeletal muscle. We are aware that the calculated sample size can be not appropriate for a full description of the morphological changes.

In the discussion section, we clarified that the low number of mice limited the investigation of secondary endpoints. Future investigations having as primary endpoint the evaluation of the molecular pathways associated with this disfunction will help to quantify the histological damages that we reported in this work.